# Towards a Survival-Based Cellular Assay for the Selection of Protease Inhibitors in *Escherichia coli*

**DOI:** 10.3390/biotech14010016

**Published:** 2025-03-07

**Authors:** William Y. Oyadomari, Elizangela A. Carvalho, Gabriel E. Machado, Ana Júlia O. Machado, Gabriel S. Santos, Marcelo Marcondes, Vitor Oliveira

**Affiliations:** 1Department of Biophysics, Escola Paulista de Medicina (EPM), Universidade Federal de São Paulo (UNIFESP), São Paulo 04021-001, Brazil; william.yoshio@unifesp.br (W.Y.O.); contatoelicarvalho@gmail.com (E.A.C.); marcelo.marcondes@unifesp.br (M.M.); 2Enzymology Laboratory, 7th, Edifício de Pesquisa 2, Rua Pedro de Toledo 669, São Paulo 04039-032, Brazil

**Keywords:** protease inhibitor, cell-based assay, high-content selection method, high-throughput screening

## Abstract

We describe a method tailored to the in-cell selection of protease inhibitors. In this method, a target protease is co-expressed with a selective substrate, the product of which kills host cells. Therefore, the method can be applied to identify potential inhibitors based on cell host survival when inhibition of the target protease occurs. The TEV protease was chosen for this proof-of-concept experiment. The genetically encoded selective substrate is a single polypeptide chain composed of three parts: (1) a ccdB protein, which can cause host cell death when it accumulates inside the cell; (2) a protease cleavage sequence that can be changed according to the target protease, in this case the TEV substrate ENLYFQ↓G (↓-predicted cleavage site); and (3) the ssrA sequence (AANDENYALAA), which drives the polypeptide to degradation by the ClpX/ClpP complex inside host *E. coli* cells. In our experiment, co-expression of the active TEV protease and this selective substrate (ccdB-ENLYFQG-ssrA) caused the death of a significant host cell population, while control assays with an inactive mutant TEV Asp81Asn did not. Details of the methodology used are given, providing the basis for the application of similar systems for other proteases of interest.

## 1. Introduction

Peptidases (proteases or proteolytic enzymes) play essential roles in several biological processes, including many pathophysiological events [1,2,3]. Peptidases are also commonly found acting as virulence factors for infectious microorganisms, viruses, and parasites, and consequently, these enzymes are potential drug targets [4,5,6,7,8]. The development of specific substrates and/or protease inhibitors is traditionally a laborious and long process [8,9,10]; therefore, methods aiming to facilitate this route have been the goal of many researchers. Among these methods, the introduction of libraries of substrates and inhibitors certainly deserves to be highlighted [11,12,13,14,15]. In-cell methods that circumvent the need to obtain pure proteins can also facilitate the search for protease inhibitors, especially for hard-to-obtain active proteases. Crucially, in-cell methods that can be coupled with libraries and that can be transformed into high-content assays are highly desired [15,16,17].

It has been demonstrated that genetically encoded substrate libraries co-expressed with a target peptidase within *E. coli* host cells can be used to develop highly susceptible substrate sequences and provide many insights into the substrate specificity of the target protease [18,19]. These genetically encoded substrate libraries are composed of GFP (green fluorescence protein) or CAT (chloramphenicol acetyl transferase) reporter proteins labeled with a peptidase-removable degron ssrA tag at the C-terminus, having the general structure of GFP-XXXXXX-ssrA or CAT-XXXXXX-ssrA (X = any amino acid residue). It has been shown that, upon controlled expression, the ssrA sequence leads this entire reporter protein to immediate degradation by the ClpXP complex inside *E. coli* cells. However, when these selective substrate libraries were co-expressed together with an active TEV protease, “positive” cells (that is, cells containing TEVpro-susceptible XXXXXX sequences) were selected (1) by green fluorescence, due to GFP accumulation [18], or (2) by survival in a chloramphenicol-containing medium, due to CAT accumulation [19], because of the removal of the ssrA degron tag by TEV protease action. Further DNA sequencing of the XXXXXX coding region revealed the sequences most susceptible to hydrolysis by TEV protease. Based on these previous works, we describe a similar in-cell system that is also based on the expression of a selective substrate with a removable ssrA tag by a co-expressed target protease. However, our method was modified to select potential protease inhibitors instead of substrates, which was done by replacing the GFP or CAT in the selective substrate for a toxic protein, the controller of cell death B (ccdB) protein (Uniprot Q52042). Consequently, the selective substrate of this modified system has the general structure ccdB-cleavage susceptible peptide linker-ssrA, and the resulting designed system has a unique feature: cleavage of the selective substrate at the peptide linker between the ccdB and the degron ssrA tag by the co-expressed target protease leads to the *E. coli* host cell’s death due to the intracellular accumulation of a toxic product (ccdB), while the full substrate (uncleaved) is degraded by the ClpXP complex. Thus, this removable ssrA tag substrate can be applied to identify “positive cells” where the inhibition of the target protease occurs, discriminating from “negative cells” where the target protease is not inhibited (active). “Positive” selection of the desired host cells (with the protease-inhibiting clone) is more appropriate and maybe even essential, since cell survival is the method of selection (positive cells must survive).

In the present work, we describe the construction of this system based on ccdB tagged with a protease-removable ssrA, and the results of cell survival assays performed with the tobacco etch virus (TEV) protease (TEVpro). TEV protease was used as a proof-of-concept for similar works, which developed methods to identify intracellular substrate specificity as mentioned above [18,19], and was therefore the ideal initial target for the development proposed.

## 2. Materials and Methods

### 2.1. Reagents

The plasmid pASK-IBA-3plus vector was purchased from IBA Lifesciences (Göttingen, Germany). Synthetic genes and primers for the point mutation and sequencing steps were purchased from Genone (Rio de Janeiro, Brazil). All other reagents were purchased from Sigma (St. Louis, MO, USA).

### 2.2. Bacterial Strain

The *Escherichia coli* strain TOP10 (F-mcrA (mrr-hsdRMS-mcrBC) 80lacZ M15 lacX74 recA1 ara 139 (ara-leu)7697 galU galK rpsL (StrR) endA1 nupG) was used as a host for the construction of the plasmids and for all cell survival assays. Luria Bertani (LB) medium was used for bacterial growth, and the appropriate antibiotic was added when needed.

### 2.3. Plasmids

The plasmid TEVpro vector constructed and used in this work has the following main characteristics: pColA ori provides ampicillin resistance and contains a gene encoding the TEV protease under an arabinose-sensitive promoter (Ara promoter) as well as a gene encoding a polypeptide sequence corresponding to the selective substrate ccdB-ENLYFQG-ssrA under an anhydrotetracycline-sensitive promoter (Tet promoter). The vector map is shown in the Appendix A, as well as the amino acid sequences of the polypeptide chains encoded by the cloned genes (Appendix A). Briefly, the pASK-IBA-3plus vector was the initial backbone, which was subsequently modified to obtain the TEVpro vector. All cloning steps were made by restriction enzyme-free PCR-based cloning methods: either CPEC [20] or TEDA [21]. First, the TEVpro vector with the pBR322 origin was obtained. This was done by inserting synthetic genes into pASK-IBA-3plus. Finally, the pBR322 origin of this vector was replaced by the pColA origin, which was amplified by PCR using pColADuet-1 (Merck, Millipore, MA, USA) as a template. All constructions were verified by DNA sequencing. The vector containing the inactive TEV was obtained by site-directed mutagenesis by PCR with specific primers (Appendix A) using the respective TEVpro plasmid as templates (Appendix A). Point mutations were confirmed by DNA sequencing and identified as TEVproi (TEVpro Asp81Asn). The plasmids are available upon request.

### 2.4. Cell Survival Assays

Plasmids containing active TEVpro or inactive (TEVproi; Asp81Asn mutation), were separately transformed into the competent *E. coli* strain TOP10 and plated on LB agar containing 100 μg/mL of ampicillin. One colony isolated from each plate was added to 5 mL of LB in a 15 mL conical tube containing 100 μg/mL of ampicillin and then grown at 37 °C at 180 RPM for 16–18 h.

For each tube, the following steps were performed: 1 mL of this medium was inoculated into 19 mL of LB (without the antibiotic) in a 50 mL conical tube and placed at 180 RPM in 37 °C until OD600 = 0.3, then the samples were separated into 4 conical tubes of 15 mL (with 3 mL each). In tube 1, nothing was added (no induction, N.I.), and it was stirred at 180 RPM in 30 °C for 2.5 h. In tube 2, 20 mM of L-(+)-Arabinose (Ara) was added for the expression of TEVpro or TEVproi, and this was stirred at 180 RPM in 30 °C for 2 h. Tube 3 was shaken for another 2 h at 180 RPM in 30 °C, then 2 ng/mL (4.32 nM) or 200 ng/mL (432 nM) of anhydrotetracycline (Tet) was added for the expression of the ccdB-ENLYFQG-ssrA complex, which was then stirred again at 180 RPM in 30 °C for 30 min. In tube 4, 20 mM of Ara was added, and after 2 h, 2 ng/mL or 200 ng/mL of Tet was added and then stirred for another 30 min at 180 RPM in 30 °C for the expression of both the enzyme and the ccdB-ENLYFQG-ssrA substrate. At the end of this period, all samples were plated on LB agar plates containing 100 μg/mL of ampicillin after successive dilutions of the bacterial culture to 101, 103, 105, and 107. The plates were kept at 37 °C for 16–18 h, and then the plate with fewer colonies (normally between 10 and 200) was used for manual counting, then multiplied by the dilution factor (101, 103, 105, and 107) of the culture media before plating. In the assays with glucose, the concentration of this compound in the LB medium was 110 mM. All assays were performed in triplicate.

## 3. Results

Assays testing the functionality of the designed system were performed under many different conditions, and the main results are summarized in the following sections. For these cell survival assays, *E. coli* TOP10 competent cells were transformed with either the TEVpro vector or TEVproi vector. Anhydrotetracycline addition induced the expression of the selective TEV substrate ccdB-ENLYFQG-ssrA, and arabinose induced the expression of either the active TEV protease (TEVpro) or TEV Asp81Asn mutant, which lacks enzymatic activity (TEVproi). It is noteworthy that active TEVpro also contains the S219V mutation (Appendix A), which prevents the auto-degradation of TEV protease, thus improving the stability of this enzyme [22].

The *E. coli* TOP10 strain was chosen because it does not metabolize arabinose, maintaining a constant level of expression throughout the experiment.

Figure 1 illustrates this assay, and Figure 2 represents a typical set of agar plates made to count the number of colonies that survived after the different treatments: no induction (N.I.); induction with only anhydrotetracycline (Tet); induction with only arabinose (Ara); induction with both anhydrotetracycline and arabinose (Ara + Tet).

Figure 2 shows that the lowest colony count on plates (highest amount of cell death) occurred in the Tet + Ara inductions in the culture samples expressing active TEVpro. The comparison of the number of colonies resulting from the assays with the inactive Asp81Asn mutant (TEVproi) clearly shows the action of TEVpro on the selective substrate (Figure 3; TEVpro versus TEVproi; Tet + Ara condition). This observation confirmed the general functionality of the designed system: the controlled expression of the selective substrate ccdB-ENLYFQG-ssrA and the action of TEV protease, which releases the ccdB toxin from the ssrA degron, causes the accumulation of ccdB toxin inside *E. coli* cells, leading to the death of these host cells. It is important to mention that under some conditions, similar lower cell counts were observed in the samples expressing only the selective substrate ccdB-ENLYFQG-ssrA (Figure 3; Tet-only). Further observations indicated that a basal leakage of the arabinose-sensitive promoter was likely occurring and causing this problem. To decrease this basal expression of TEVpro due to Ara promoter leakage, glucose was added to the medium during the cell culture growth and induction periods, and the results were compared with the glucose-free assays (also shown in Figure 3). The consequent increase in cell survival observed in these assays in the presence of glucose, especially the increase observed in the sample induced with Tet-only, confirmed the leakage of TEVpro (Ara promoter) in the assays without glucose. In the Ara-only inductions, the assays with TEVi expression also showed that the TEV protease activity within cells may have some toxicity and/or that the Tet promoter also presents some leakage. Therefore, in addition to control of the Ara promoter, which can be done by adding glucose to the culture medium, we also had to investigate conditions that could decrease the expression of the ccdB-ENLYFQG-ssrA substrate, which was done by decreasing the concentration of anhydrotetracycline added for induction (Figure 3). Better conditions were indeed observed with less anhydrotetracycline (Figure 3; Tet 2 ng/mL). Finally, by controlling Ara promoter leakage with glucose and reducing the expression of the ccdB-ENLYFQG-ssrA substrate by decreasing the anhydrotetracycline concentration 100-fold (relative to the concentrations used for regular protein expression, which is 200 ng/mL), a very significant difference of more than three to four orders of magnitude in colony counts could be observed, when comparing the results obtained with TEVproi and TEVpro (Figure 3; Ara + Tet inductions, cyan bars).

## 4. Discussion

The fundamental basis of the methodology described here is the co-expression within the same host cell of a target protease and its selective substrate, the product of which kills the host cell (Figure 1). These cells can then be used to identify potential inhibitors from intracellular inhibitor libraries, such as cyclic peptide SICLOPPS libraries [23,24], for example, based on cell survival when inhibition of the target protease occurs. The enzyme–substrate pair selected for this proof-of-principle work was TEV protease and the selective substrate ccdB-ENLYFQG-ssrA.

As stated in the introductory section, TEV protease was selected because it was successfully tested in similar in-cell-based systems [18,19], where it was verified that TEVpro efficiently removes the ssrA degron tag from the genetically encoded substrates: GFP-ENLYFQG-ssrA and CAT-ENLYFQG-ssrA. Still, cleavage of these substrates by TEVpro likely occurred at the ENLYFQ↓G site, since it is well known that TEVpro recognizes the sequence ENLYFQG (TEV site) with high specificity, promoting the cleavage of the Q-G peptide bond [25]. Consequently, based on these previous observations, the following substrate was designed for the in-cell “positive” selection desired in the present experiment: ccdB-ENLYFQG-ssrA. For other target proteases, it might be necessary to first develop efficient intracellular substrates for the target peptidase by using the system described in Löfblom and Samuelson’s works [18,19] to subsequently change the substrate selection marker from GFP or CAT to the toxic ccdB, and then to use “positive” selection, which is enabled by this change, to select inhibitors.

To facilitate the coupling of this selection method with a genetically encoded inhibitor library, we chose to place all the selective elements in a single plasmid. Therefore, a two-promoter plasmid vector was constructed (Appendix A). A gene containing the coding sequence of this polypeptide ccdB-ENLYFQG-ssrA was synthesized and cloned under the control of an anhydrotetracycline-sensitive promoter (the Tet promoter), and the TEVpro coding gene sequence was cloned under the control of the Ara promoter.

In fact, the gene under the arabinose promoter’s control encodes the following fusion protein: MBP-ENLYFQG-TEVpro (MBP—maltose binding protein) (Appendix A). Expression of this fusion protein results in much higher TEVpro activity in the intracellular environment of host *E. coli* cells compared to the expression of TEVpro without this fusion protein at its N-terminal. This is likely because MBP helps increase the solubility of TEVpro and, therefore, its solubility and activity in the intracellular environment [25]. Although, interestingly, it can be removed by TEVpro’s own activity after expression, without a significant loss in TEVpro solubility [18,19,25]. Indeed, we observed by SDS-PAGE analysis that most of the expressed MBP-ENLYFQG-TEVpro was auto-processed, therefore showing that the fusion protein was hydrolyzed at the TEV cleavage site between the MBP and TEV sequences (Appendix A).

In addition to SDS-PAGE analysis, to test the correct and independent expression of both genes, a vector plasmid was constructed in which the ccdB-ENLYFQG-ssrA and TEVpro genes were replaced by fluorescent protein genes: the ccdB-ENLYFQG-ssrA substrate was replaced with mClover3 and TEVpro was replaced with mRuby3 (Appendix A). Subsequent cultures were made with *E. coli* TOP10 cells transformed with this plasmid, where mClover3 expression was under the control of the Tet promoter and mRuby3 expression was controlled by the Ara promoter. Afterwards, fluorescence measurements of diluted cell extracts from these small-scale cultures, in the presence or in the absence of anhydrotetracycline and/or arabinose, confirmed the correct expression of the fluorescent green (mClover3) and/or red (mRuby3) proteins (Appendix A). It was also possible to estimate the expression ratio based on fluorescence measurements normalized by mClover3 or mRuby3 brightness [26], which was approximately 11:1 (Tet-only: Ara-only) (Appendix A).

Based on this mClover3/mRuby3 ratio, we can assume that the ratio of ccdB-ENLYFQG-ssrA/TEVpro is approximately the same at the end of the induction time in the described assays (i.e., 11 molecules of substrate for every enzyme). This supports the choice of a stronger promoter for the substrate (the Tet promoter) and a weaker promoter for the enzyme the (Ara promoter) in order to have more substrate than enzyme expressed within the host cells (~10:1). Finally, these fluorescent measurements also showed that when both inductions (Tet + Ara) occurred at the same time, the amount of both proteins expressed was lower than the sum of the individual proteins when they were expressed separately). Although these differences were not so substantial, such observations led us to introduce an additional control in which the expressed TEV protease is inactive, in addition to the Tet-only and Ara-only conditions. For this purpose, a site-directed mutation was introduced by PCR, with specifically designed primers (Appendix A), at the TEV gene, generating a plasmid that encoded a mutant TEVpro containing an asparagine residue at position 81, replacing the original aspartic acid residue at the active TEVpro. This mutant TEV Asp81Asn protease (TEVproi) had no enzymatic activity [22]. Assays with an inactive mutant enzyme as a control should be essential when using this method for the selection of inhibitors in order to eliminate false positives, especially inhibitors of the ClpX/ClpP degradation complex.

But it is noteworthy that the individual expression of the selective substrate (Tet-only) or of the target enzyme (Ara-only) are also essential controls for these assays. For instance, in the results of the Tet-only assays, we observed that the finely controlled expression of the selective substrate was essential to find the correct induction level, which could be counterbalanced by the degradation of ccdB-ENLYFQG-ssrA by the ClpX/ClpP complex: i.e., resulting in no detection of death for a significant host cell population in the cell survival assays. Since it could be observed that high levels of expression (200 ng/mL of anhydrotetracycline) exceeded the capacity of the ClpX/ClpP complex to degrade all the ccdB-ENLYFQG-ssrA produced, which caused the death of a significant population of host cells even without the induction of TEVpro expression and even after the suppression of Ara promoter leakage by the addition of glucose to the culture medium during the survival assays (Figure 3). The leakage of the Ara promoter was only detectable due to the Tet-only controls. Additionally, it is crucial to highlight that changing the replication origin of the pASK-IBA-3plus vector from the high-copy pBR322 to the low-copy pColA was essential for reducing the expression of ccdB-ENLYFQG-ssrA to a level where minimal cell death was observed in the control assays (Tet-only conditions). With the TEVpro vector, which uses the high-copy pBR322 origin, we could not identify a concentration of anhydrotetracycline that avoided significant cell death in the Tet-only controls.

Taken together, the results obtained showed that by controlling the expression conditions of the protease and its selective substrate, a very large difference (three to four orders of magnitude) could be observed in the cell survival assays between the assays with the active TEV and the inactive TEV mutant (Figure 2 and Figure 3). This demonstrated the functionality of the proposed selection method.

However, we need to emphasize two important limitations of this technique. The first is that the protein must be soluble in the intracellular environment; that is, it does not aggregate (e.g., due to the high amount of hydrophobic residues)—as we observed in tests with the dengue protease (NS2B/NS3 protease)—although, in many cases, we can add a fusion protein to increase solubility, like we did with MBP fused with TEVpro. The second limitation is that the protein must not have post-translational modifications important for its activity, such as disulfide bonds between two cysteines, as the bacteria do not have the necessary organelles to make these modifications.

In general, these are the same limitations that apply to the expression of recombinant proteins in bacteria, which is a methodology widely used in the community. In this way, we believe that many proteins expressed in bacteria can be applied normally in this method.

## Figures and Tables

**Figure 1 biotech-14-00016-f001:**
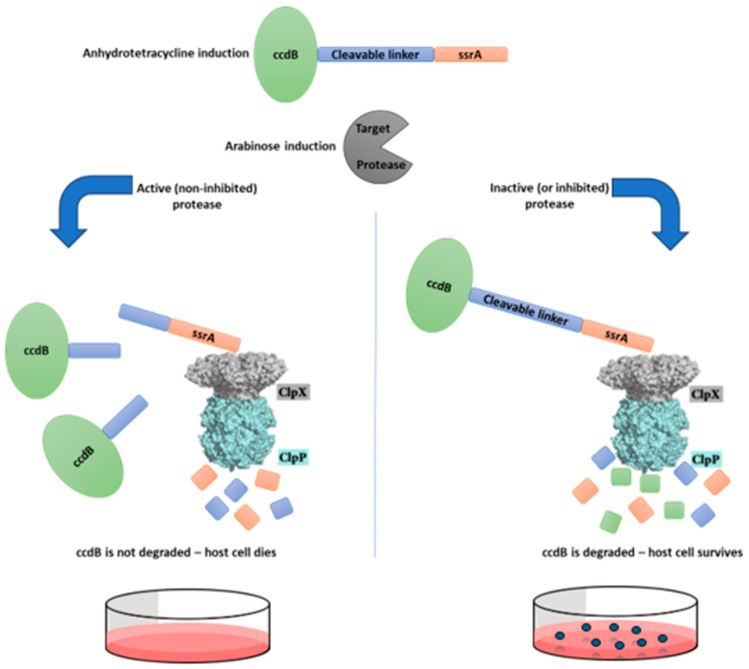
**Schematic representation of the survival-based cellular assay disegned for selection of protease inhibitors.** This system is based in a selective substrate containing a toxic protein linked to the ssrA degron sequence by a protease-removable peptide linker. This substrate, when hydrolyzed by the target protease, generates a ccdB-peptide fragment, which is not degraded by the ClpXP complex due to the removal of the ssrA degron tag, and therefore the host cell does not survive. However, if the target protease is inhibited and does not cleave this substrate, the host cell should survive, as the ssrA sequence will direct the entire substrate to be degraded by the ClpXP complex.

**Figure 2 biotech-14-00016-f002:**
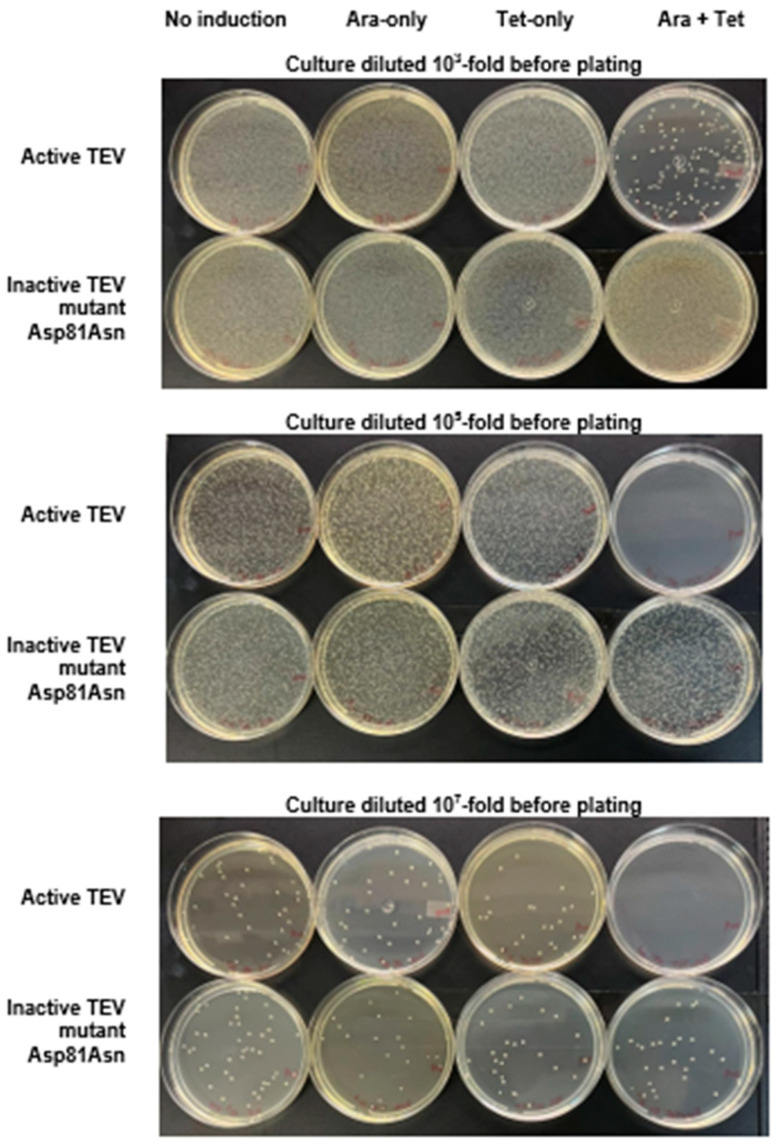
**Cell survival assay with the selective substrate ccdB-ENLYFQG-ssrA expressed under the control of the Tet promoter and TEVpro (or TEVproi) under the control of the Ara promoter.** The photographs show typical results of culture plating of *E. coli* TOP10 transformed with the TEVpro vector (upper plates at each level of dilution) or TEVproi (lower plates at each level of dilution), after the following treatments (as described Section 2.4): no induction (N.I.); induction with only arabinose (Ara); induction with only anhydrotetracycline (Tet); induction with both anhydrotetracycline and arabinose (Tet + Ara). The photos show plates from one of the assay triplicates (condition: Tet 2 ng/mL with glucose), whose comparative quantification with other assay conditions if represented in Figure 3 by the cyan bars.

**Figure 3 biotech-14-00016-f003:**
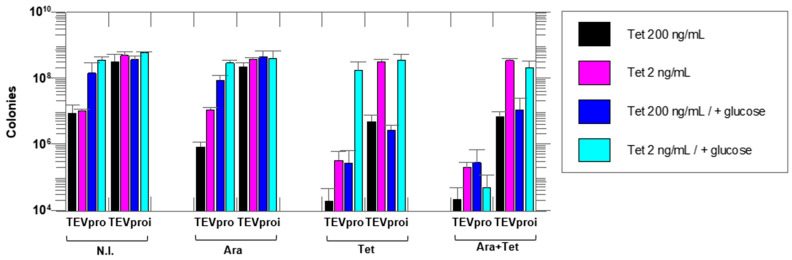
**Cell survival assays with the selective substrate ccdB-ENLYFQG-ssrA expressed under the control of the Tet promoter and TEVpro (or TEVproi) under the control of the Ara promoter.** The number of colonies on agar plates after growth of an *E. coli* TOP10 cell culture containing the TEVpro vector (TEV) or TEVproi vector (TEVi). The expression inducers added to the different samples were: N.I.—no induction; Ara—arabinose; Tet—anhydrotetracycline and Ara + Tet—arabinose and anhydrotetracycline. Addition of glucose suppresses the Ara promoter by decreasing the basal expression (leakage) of the TEV protease that occurs even in the absence of arabinose in the culture medium. Different concentrations of anhydrotetracycline were tested, as shown in the figure. All assays were performed in triplicate, and all differences were statistically significant in the Ara + Tet group, comparing the TEVpro with the TEVproi groups. Data represents the mean of triplicate assays plus or minus the standard deviation. Statistical analyses were carried out using GraphPad Prism 8.

## Data Availability

The original contributions presented in this study are included in the article/Appendix A. Further inquiries can be directed to the corresponding author(s).

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
