# Peer review of "Towards a Survival-Based Cellular Assay for the Selection of Protease Inhibitors in Escherichia coli"

_biotech, 2025, doi:10.3390/biotech14010016_

Round 1

Reviewer 1 Report

Comments and Suggestions for Authors

Dear Authors,

In this research, the authors, William Y. Oyadomari et al., investigated the cell survival assay for the rapid selection of protease inhibitors. The authors thoroughly document this research as an initial exploration of selected inhibitors in cells. This approach can certainly save time compared to traditional methods for selecting inhibitors. Overall, the study is commendable; however, I have a few minor remarks for the authors. These remarks are intended to enhance the clarity and impact of the findings presented in this important work.

 Review Comments:

 1.       Evaluation of Alternative Proteases:

In this study, TEV protease was selected for the proof-of-concept experiment. Have the authors considered evaluating alternative proteases to determine the broader applicability of this method? Expanding the investigation to include other proteases could provide valuable insights into the versatility and adaptability of the proposed selection approach. Such exploration may enhance the general utility of the methodology in various biological contexts and increase its relevance to a wider audience.

 2.       Discussion on Limitations of TEV Protease Exclusivity:

While TEV protease serves as an excellent proof-of-concept, it would be beneficial for the authors to include a discussion on the limitations of focusing exclusively on this protease. Highlighting any constraints, such as its specific characteristics or potential challenges in applying the method to other proteases, would provide a balanced perspective. This discussion could also offer guidance for future research and facilitate the broader application of the methodology in protease inhibitor selection.

3.       System Flexibility for Endopeptidases vs. Exopeptidases:

Is the described system designed solely for endopeptidases, or could it also be adapted for exopeptidases? Investigating the potential for exopeptidase utilization would broaden the scope of this research and demonstrate the flexibility of the system. Exploring a wider range of proteases, including both endopeptidases and exopeptidases, would significantly enhance the relevance of the findings. Furthermore, this analysis could showcase the robustness of the methodology and its potential applications in diverse enzymatic and biochemical contexts.

4.       Expansion of Protease Range:

To validate the durability and applicability of the proposed technique, the authors may consider expanding their experiments to include a broader range of proteases. This would help determine whether the system can accommodate proteases with different cleavage specificities, which may improve its utility in varied research scenarios. Such an expansion would also provide a more comprehensive understanding of the technique's strengths and limitations.

Reviewer 2 Report

Comments and Suggestions for Authors

Authors describe a study in which they generate a specific plasmid to be used for the selection of protease inhibitors in E.Coli. This is a proof-of-principle study. The method described by authors allow the selection of inhibitors of a protease under study by evaluating the survival rate of plated E.Coli.

One great limit of this research is that only one protease was tested as proof of principle, and only one genetically encoded inactive protease have been evaluated. Thus, in my opinion, to prove the claim that the method allow the “rapid selection of potential protease inhibitors” it is necessary to test additional molecules known to inhibit (partially or completely) TEV protease.

Moreover, in the end of the discussion section a lot of limitations and potential issues related to the presented method are highlighted, thus pointing out that it will be difficult to be applied.

In its present form the paper suffers from gaps in its presentation and content.

Here are listed some additional considerations to improve the quality of the paper:

Lines 58-59: I absolutely do not agree with this sentence: "However, our method has been modified to a rapid selection of potential protease inhibitors instead of substrates". In my opinion this method is far from being rapid, indeed the induction, selection, dilution and plating require few days of work, but the generation of the plasmid specific for the protease to be tested seems very far from being rapid.

Materials and methods: no information are provided about how many replicates were done for each condition tested.

Figure 3. How data about colonies are reported in the figure (mean±standard deviation)? Were some statistical tests applied? Nothing about these data is reported in materials and methods.

Figure 3S. It is not clear to me how authors identified bands on the basis of a Coomassie staining. Please perform additional experiments to support your claims.

Lines 213-227: This paragraph is not very clear to me. There are a lot of additional information regarding methods and results that are presented in the discussion. I think this is confusing for readers. How these changes to the plasmid will affect the method and how can it be improved needs to be discussed.

In my opinion the presented method is not easy to apply to proteases' inhibitors screening

Comments on the Quality of English Language

Some sentences can be shortened or modified to improve the quality of presentation

Reviewer 3 Report

Comments and Suggestions for Authors

1.    The title is clear but could be more specific by mentioning the application scope (e.g., "Towards a Survival-Based Cellular Assay for Rapid Selection of Protease Inhibitors in E. coli").

2.    Consider linking the introduction more explicitly to potential real-world applications, such as drug development.

3.    Include more detailed diagrams or visual aids for the experimental setup to enhance comprehension.

4.    Specify reagents and their sources.

5.    Ensure clarity when describing complex procedures like plasmid construction or induction conditions. Simpler flowcharts might help.

6.    Ensure all citations are formatted consistently, especially regarding journal names and DOI inclusion. Frontiers-style formatting should be verified. And add some more relevant references like,

 Ø  Chouhan M, Tiwari PK, Mishra R, Gupta S, Kumar M, Almuqri EA, Ibrahim NA, Basher NS, Chaudhary AA, Dwivedi VD, Verma D, Kumar S (2024). Unearthing phytochemicals as natural inhibitors for pantothenate synthetase in Mycobacterium tuberculosis: A computational approach. Frontiers in Pharmacology. 15: 1403900.

 Ø  Tiwari PK, Chouhan M, Mishra R, Gupta S, Chaudhary AA, Zharani MA, Qurtam AA, Nasr FA, Jha NK, Pant K, Kumar M, Kumar S (2024). Structure-based virtual screening methods for the identification of novel phytochemical inhibitors targeting furin protease for the management of COVID-19. Frontiers in Cellular and Infection Microbiology. 14: 1391288.

Reviewer 4 Report

Comments and Suggestions for Authors

Comments:

The first comment regards the methodology: The authors chose the TEV protease. However, I guess that it is important to test whether this kind of assay works well also with mammalian protease.  I mean, are there any information regarding the activity of mammalian proteases expressed in bacteria? Could authors add any comment on it?

2- Into figure 3 lacks the statistics.  Also, from which plate colonies were counted? Please indicate it. And how many time this count was repeated? Also, it lacks the method with which authors count colonies (automatically?). Please add software used.

figure 3s_ It is important to assess by western blot that the bands indicated by arrows are the proteins supposed to be. therwise you can cut the band and analyse by MS/MS.

Figure 4s. Also, to confirm that cells are fluorescent, check the culture's fluorescence under a microscope, by excitating at the proper wavelength (add images in the paper).

Finally , in order to have an easy protocol to test different inhibitors , I suggest to measure the OD600 of cultured bacteria in a multi-well plate reader. In a such a way, people can read the growth of bacteria and test at the same time different inhibitors.

Round 2

Reviewer 4 Report

Comments and Suggestions for Authors

Now the paper is improved and can be published.